# Impact of COVID-19 Lockdown, during the Two Waves, on Drug Use and Emergency Department Access in People with Epilepsy: An Interrupted Time-Series Analysis

**DOI:** 10.3390/ijerph182413253

**Published:** 2021-12-16

**Authors:** Ippazio Cosimo Antonazzo, Carla Fornari, Sandy Maumus-Robert, Eleonora Cei, Olga Paoletti, Sara Conti, Paolo Angelo Cortesi, Lorenzo Giovanni Mantovani, Rosa Gini, Giampiero Mazzaglia

**Affiliations:** 1Research Centre on Public Health (CESP), University of Milan-Bicocca, 20900 Monza, Italy; ippazio.antonazzo@unimib.it (I.C.A.); e.cei@campus.unimib.it (E.C.); sara.conti@unimib.it (S.C.); paolo.cortesi@unimib.it (P.A.C.); lorenzo.mantovani@unimib.it (L.G.M.); giampiero.mazzaglia@unimib.it (G.M.); 2Team Pharmacoepidemiology, Bordeaux Population Health Research Center, Inserm U1219, University of Bordeaux, 33000 Bordeaux, France; sandy.robert@u-bordeaux.fr; 3Epidemiology Unit, Regional Agency for Healthcare Services of Tuscany, 50141 Florence, Italy; olga.paoletti@ars.toscana.it (O.P.); rosa.gini@ars.toscana.it (R.G.); 4Value-Based Healthcare Unit, IRCCS MultiMedica, 20099 Sesto San Giovanni, Italy

**Keywords:** COVID-19, Italy, epilepsy, anti-seizure medication, emergency department access, time-series analysis

## Abstract

Background: In 2020, during the COVID-19 pandemic, Italy implemented two national lockdowns aimed at reducing virus transmission. We assessed whether these lockdowns affected anti-seizure medication (ASM) use and epilepsy-related access to emergency departments (ED) in the general population. Methods: We performed a population-based study using the healthcare administrative database of Tuscany. We defined the weekly time series of prevalence and incidence of ASM, along with the incidence of epilepsy-related ED access from 1 January 2018 to 27 December 2020 in the general population. An interrupted time-series analysis was used to assess the effect of lockdowns on the observed outcomes. Results: Compared to pre-lockdown, we observed a relevant reduction of ASM incidence (0.65; 95% Confidence Intervals: 0.59–0.72) and ED access (0.72; 0.64–0.82), and a slight decrease of ASM prevalence (0.95; 0.94–0.96). During the post-lockdown the ASM incidence reported higher values compared to pre-lockdown, whereas ASM prevalence and ED access remained lower. Results also indicate a lower impact of the second lockdown for both ASM prevalence (0.97; 0.96–0.98) and incidence (0.89; 0.80–0.99). Conclusion: The lockdowns implemented during the COVID-19 outbreaks significantly affected ASM use and epilepsy-related ED access. The potential consequences of these phenomenon are still unknown, although an increased incidence of epilepsy-related symptoms after the first lockdown has been observed. These findings emphasize the need of ensuring continuous care of epileptic patients in stressful conditions such as the COVID-19 pandemic.

## 1. Introduction

The coronavirus disease 2019 (COVID-19) pandemic has affected everyone around the world, not least through pressure on healthcare systems and delivery of care. The initial wave and consequent lockdowns led to the cancellation of routine investigations and elective interventions, and many health care providers had to move to alternative models of care delivery [1,2,3,4]. The new disease started in China in late December 2019 and rapidly spread worldwide, becoming a global pandemic issue [1]. Italy was the one of the earliest affected European countries, with the highest number of confirmed COVID-19 cases. The rapidly increasing number of confirmed cases led the Italian government to impose, from 9 March 2020 to 15 June 2020, its first nation-wide lockdown to limit the virus diffusion in the general population [5]. Owing to a second COVID-19 wave, a new lockdown was implemented from 16 November 2020 [5]. The lockdowns forced the general population to stay at home and to go out only for vital needs. On the other hand, an urgent reorganization of the healthcare delivery services, with prioritization of COVID-19 cases, was necessary. This had the potential to affect the management of patients with chronic conditions that needed continuous monitoring, such as patients with epilepsy [6].

Epilepsy is one of the most common neurological conditions, impacting about 46 million people worldwide [7]. This disease affects people of all ages and requires long-term and sometime lifelong treatment [7]. Stressful events, such as natural disasters and epidemics, have been associated with epilepsy onset and worsening, as well as low patient empowerment [8,9,10].

Some studies have already analyzed the impact of the COVID-19 outbreak on patients with epilepsy. However, these studies were mainly questionnaire-based [11] or carried out in a small sample focusing on assessing either the risk of relapses [6,12,13] or anti-seizure medication (ASM) adherence during the first COVID-19 outbreak [14]. To the best of our knowledge, information on ASM use and emergency department (ED) access for epilepsy during the whole of the COVID-19 waves is still scarce and deserves careful evaluation. This study, therefore, aimed at evaluating the impact of the COVID-19 pandemic on incidence and prevalence of ASM use and ED access due to epilepsy and related disorders in the general population.

## 2. Materials and Methods

### 2.1. Study Design, Data Source and Ethical Approval

This is an observational study based on the healthcare administrative database (HAD) of Tuscany. This source of data has already been used to conduct a study on ASM use [15]. The HAD contains data on healthcare services accessed in the regional area, reimbursed by the Italian Healthcare System (NHS) to all regional citizens, and the demographic registry.

Encrypted demographic information (i.e., dates of birth/enrollment and death/disenrollment) from the demographic registry was linked, at patient level, with the outpatient pharmacy claims registry and the ED access database. The pharmacy claims registry includes data on dispensing date, number of packages, substance name, anatomical therapeutic chemical code (ATC) and defined daily dose (DDD). The ED database reports information related to date of access, date of discharge/hospitalization, one main and five secondary diagnoses codified using the International Classification of Diseases, Ninth Revision, Clinical Modification (ICD-9-CM) [16]. This study was approved by the Agenzia Regionale di Sanità della Toscana Internal Governance Board.

### 2.2. Study Population and Outcome

The reference population was the general population of Tuscany, which accounted for approximately 3.7 million inhabitants. The population was stable through the study period, males accounted for 48%; 15% of the population was under 18-years and 32% was above 60-years. In the general population, we built the time series of weekly prevalence and incidence of ASM use and of weekly access to the ED for epilepsy and related disorders from 1 January 2018 to 27 December 2020 (study period).

To evaluate the use of ASM in the general population we selected a dynamic cohort of subjects with at least one dispensing of ASM (ATC: N03A*) in the study period. The date of the first ASM dispensing was considered as the index date. We excluded patients with less than 1 year of database history prior to the index. For each patient, ASM dispensing was collected from the index date until the 27 December 2020 or death/emigration, whichever came first. 

We then computed treatment episodes of drug use for each subject. For this purpose, the duration of each dispensing was estimated by dividing the total amount of active substance contained in each dispensing by the relevant Defined Daily Dose (DDD) [17,18]. We considered having a continued treatment episode if a new dispensing occurred within 30 days after the end of the previous one (grace period) [18]. Each subject with a treatment episode retrieved during the observational period was included in the computation of weekly prevalence of use for the episode’s duration. Subjects with treatment episodes with no ASM use within one year prior to the initiation of the date-of-treatment episode were included in the computation of incidence of use only for the first date-of-treatment episode.

Finally, the weekly prevalence and incidence of ASM use were computed as the number of prevalent or incident users in the corresponding week, divided by the number of inhabitants living in the region as of 1 January of each corresponding calendar year as reference population [19].

The times series of ED accesses for epilepsy and related diagnoses included all subjects who accessed an ED during the study period with the following main diagnoses: epilepsy (ICD9-CM: 345.x), myoclonus (333.2), convulsion febrile or afebrile spasm (780.3), and other abnormal involuntary movements (781.0). Similarly, we estimated the weekly ED access by dividing the number of cases occurred in a specified week during the study period with the number of inhabitants living in the region as of 1 January of each corresponding year as reference.

### 2.3. Statistical Analysis

To assess the effect of lockdown measures (LM) on study outcomes, the observation period was divided into segments: a pre-lockdown period from 1 January 2018 to 8 March 2020 (114 weeks), a first lockdown period from 9 March 2020 to 15 June 2020 (14 weeks), a post-lockdown period from 16 June 2020 to 15 November 2020 (22 weeks) and a second lockdown period from 16 November 2020 and 27 December 2020 (6 weeks).

We then used an Interrupted Time Series (ITS) approach [20,21] to estimate the effect of LM on the observed outcomes time series. This is a quasi-experimental design to evaluate longitudinal effects of time-delimited interventions accounting for seasonality and secular trend. In detail, we applied a quasi-Poisson generalized additive model [22] with the weekly count of the observed outcome as response variable (Y) and the reference population as offset variable to transform the count outcome in incidence or prevalence. The fitted model was of the form:
Log[E(Y_i_)] = β_0_ + f(week_i_) + β_1_I(holiday_i_) + β_2_I(First lockdown_i_) + β_3_(First lockdown week_i_) + β_4_I(Post-lockdown_i_) + β_5_(Post-lockdown week_i_) + β_6_I(Second lockdown_i_) + β_7_(Second lockdown week_i_).

The model predictors were a non-linear function of the week (f(week_i_), spline function) and a dummy holiday indicator (0 = no, 1 = yes) to account for time trend and seasonality. The coefficients: β_0_ represented the baseline level during the pre-lockdown period; β_2_, β_4_ and β_6_ estimated the level change during the first lockdown, the post-lockdown and the second lockdown, respectively; β_3_, β_5_ and β_7_ estimated the trends/slopes of the time series during the aforementioned periods [22,23].

In the model, a level change means an abrupt effect of the intervention, whereas a change in trend/slope represents a gradual change in the estimated outcome [22].

When analyzing the time series of ED access, the holiday indicator was substituted with a month indicator to correct for seasonality.

Moreover, we investigated a possible delayed effect of LM implementation or cancellation, using the delayed or “lagged” level and slope indicators for all segments. Statistical significance of the parameters and the goodness of fit of the model were used to choose the best model. Significance was defined as a *p*-value less than 0.05. The models were implemented separately for each outcome.

Sensitivity analysis for prevalence of drug use was conducted by using a longer grace-period (60 days) to assess potential variation in prevalence estimation during the studied period.

In this study, both the data processing and data analysis were performed using the R studio software (version 4.0.2 RStudio, PBC: Boston, MA, USA). In particular, the ITS analysis was conducted by using “tsMODEL”, “splines”, and “mgcv” R packages.

## 3. Results

### 3.1. Prevalence of AEs Use

As reported in Table 1 and Figure 1, the two lockdowns affected ASM prevalence in the general population. In detail, the ITS analysis suggested a slight reduction of weekly prevalence of ASM use during both the first and second lockdowns (Table 1 and Figure 1A). Such reduction was observed after 4 weeks from the first LM implementation (Prevalence Ratio (PR): 0.95; 95%CI: 0.94–0.96) and after 3 weeks from the second LM implementation (0.97; 0.96–0.98). The sensitivity analysis of ITS models on ASM prevalence with a grace period of 60 days confirmed the results observed in the main analysis (Table A1, Figure A1).

### 3.2. Incidence of AEs Use

The weekly incidence of ASM use reported significant variations during the study period. In detail, the ITS regression model depicted a significant reduction of ASM incidence after the first LM implementation (Incidence Ratio (IR): 0.65; 95%CI: 0.59–0.72) followed by a gradual increase from the 4th week of the first lockdown (1.04; 1.03–1.05) (Table 1 and Figure 1B) with a plateau reached at the 4th week from the end of the first lockdown. The second LM implementation led to a new decrease of the incidence of ASM use (0.89; 0.80–0.99).

### 3.3. Emergency Department Access

The analysis of weekly ED access for epilepsy and related disorders showed significant variation as already described for ASM use (Table 1 and Figure 2). In fact, the ITS analysis showed a significant reduction in the occurrence of event during the first week of the first lockdown (Events Ratio (ER): 0.72; 95%CI: 0.64–0.82). As reported in Figure 2 and Table 1, an increase in event occurrence was observed during the first week of the post-lockdown period (ER: 1.31; 1.15–1.48).

## 4. Discussion

In this large population-based study, we observed during the first LM implementation a significant reduction in both incidences of ASM use and epilepsy-related ED access compared to pre-lockdown. On the contrary, for ASM prevalence, only a slight reduction was reported.

Fear of contagion, along with the overload of the healthcare services associated with the hospital adaptation/reorganization for COVID-19 cases, might have limited non-COVID-19 patients’ access to healthcare services, including ED access for patients with epilepsy-related symptoms [6,13,24].

It is also possible to speculate that even the observed decrease of ASM incidence might be related to the corresponding decrease of epilepsy-related ED access, since lack of availability of trained clinicians and specialist services, as observed during the COVID-19 epidemic peaks, is likely to affect treatment initiation [25,26]. On the other hand, during the lockdown, telemedicine consultation experienced a rapid increase [27,28]. The use of this new tool allowed healthcare professionals to manage patients with epilepsy outside the hospital settings, thus potentially explaining why the prevalence of ASM use did not experience the same dramatic drop observed for the incidence of ASM use.

In the post-lockdown, we observed an increase in ASM use and ED access. Nonetheless, ED access never returned to the values reported in the pre-lockdown period, whereas ASM use increased, with reported values higher than those registered during the pre-lockdown. These findings suggest an increased incidence of epilepsy-related symptoms in the general population following the first epidemic peak. Moreover, it is likely that during the post-lockdown, the management of patients with epilepsy was conducted outside the hospital setting not only for prevalent but also for incident cases.

The increased incidence of ASM use in the post-lockdown may be the result of the stressful effects of the LM implementation experienced by susceptible individuals. Animal studies have in fact suggested that stressful events may trigger epilepsy onset and/or worsening. Thus far, two pathological mechanisms have been hypothesized: first, stress can increase glucocorticoid levels that influence serotonin, glutamate and GABA levels with consequent cortical hyperexcitability and seizure onset [29,30,31,32]; second, stress can activate the hypothalamic-pituitary-adrenal axis, which might be associated with an increased epileptic form activity [33,34,35,36,37]. Although some evidence suggested that stress is a risk factor for epilepsy onset also in humans, this association deserves further investigation in a non-COVID-19 context [38,39,40]. In addition, it is worth mentioning some evidence that suggested a neurotropism of SARS-CoV-2 with consequent onset of neurological symptoms and potential sequelae in COVID-19 patients such as dizziness, ataxia and seizures. Therefore, it is reasonable to speculate that a minority of the new ASM users might be treated for epilepsy and for prevention of further seizures resulting from post-acute COVID-19 [41,42].

Finally, study results indicate a significantly lower impact of the second LM in the observed outcomes. This different impact might be the effect of the stricter public health measures implemented during the first COVID-19 wave compared to the second one. This might have also contributed to the selection of more severe patients who accessed the ED during the post-lockdown and subsequent period, with consequent reduction of non-severe cases [43].

This study has some limitations that should be acknowledged. First, in the study we did not include benzodiazepines, which are commonly prescribed for epilepsy management [44,45]. This class of drugs is not reimbursed by the Italian National Health System and, therefore, benzodiazepines cannot be retrieved from the HAD flow. Second, the Italian pharmacy claim database does not include information on drug indication. Additionally, we performed a time-series analysis with the use of ecological data. This approach does not allow ensuring a causal correlation between the observed outcomes and COVID-19 outbreak, although the change in ASM use and ED access around the first lockdown period appeared evident. Finally, regional differences could limit the generalization of our findings to the whole Italian population. Moreover, different lockdown measures implemented during the two COVID-19 waves across the Italian regions and European countries might have had differing impacts on the drug use and healthcare resource utilization. Further research should explore the impact of the restrictive public health measures in European and non-European countries on the aforementioned outcomes.

## 5. Conclusions

Our findings emphasize the need to provide appropriate management of patients with epilepsy, particularly in potentially stressful conditions such as the COVID-19 pandemic. Patients with epilepsy are more prone to develop psychiatric and neurological comorbidities (such as stroke, migraine, mood and anxiety disorders and depression) compared with non-epileptic individuals [45]. Therefore, the under-diagnosis as well as the potential under-treatment that occurred during the pandemic waves might be associated with future increased risk of the occurrence of comorbidities [45]. Future studies are required to assess the mid- and long- term effects of LMs in patients with epilepsy. Finally, from a methodological point of view, this type of analysis represents a reliable method to explore whether similar trends might have occurred among individuals with other chronic conditions during the emergency situation. These results might be used by relevant stakeholders to implement new strategies to guarantee continued medical care in future public health emergencies. Additionally, evidence from these studies might be used to target specific health interventions in sub-groups of patients who have peculiar needs in order to mitigate the impact of future COVID-19 waves on their management.

## Figures and Tables

**Figure 1 ijerph-18-13253-f001:**
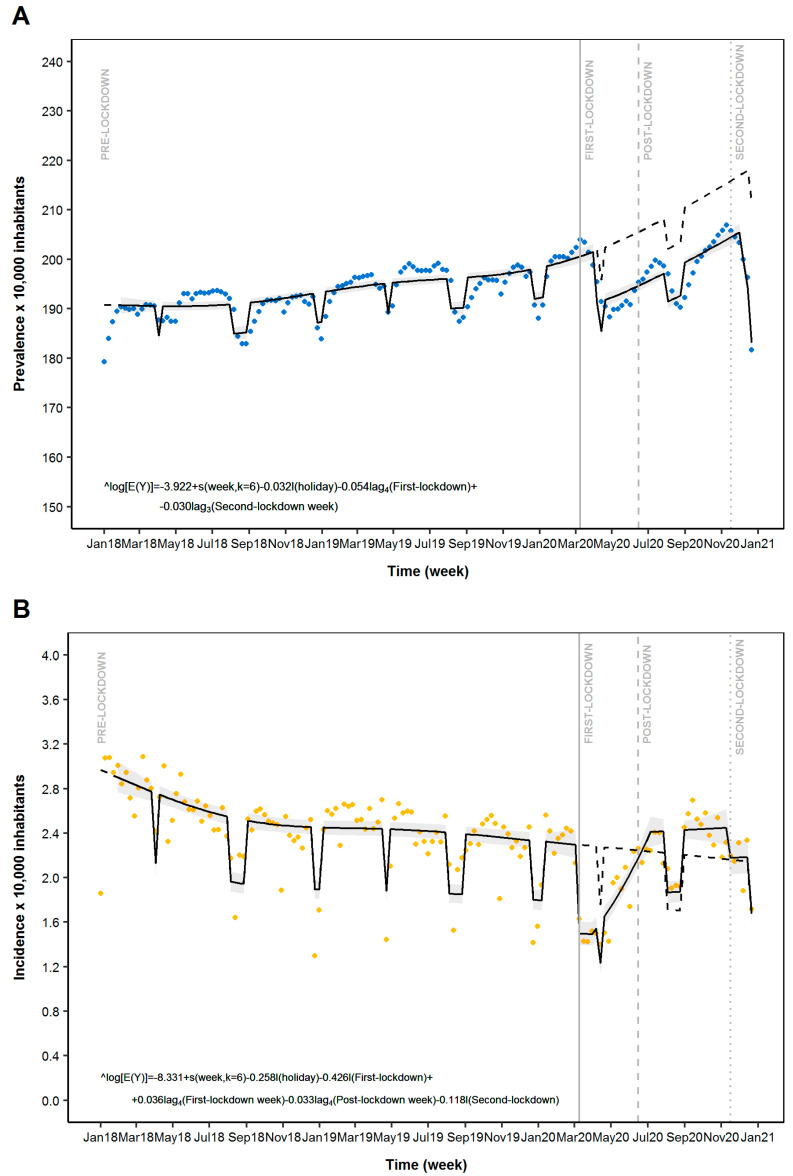
Time series analysis of anti-seizure medication (ASM) use across different periods: Pre-Lockdown, First Lockdown, Post-Lockdown and Second Lockdown. Legend: Panel (**A**) Prevalence of ASM use; Panel (**B**) Incidence of ASM use. Blue dots: estimated prevalence of ASM use; Orange dots: estimated incidence of ASM use; solid line: predicted model based on estimated outcome; Grey zone: 95%CI of the predicted model; Dashed line: expected scenario in the absence of LMI.

**Figure 2 ijerph-18-13253-f002:**
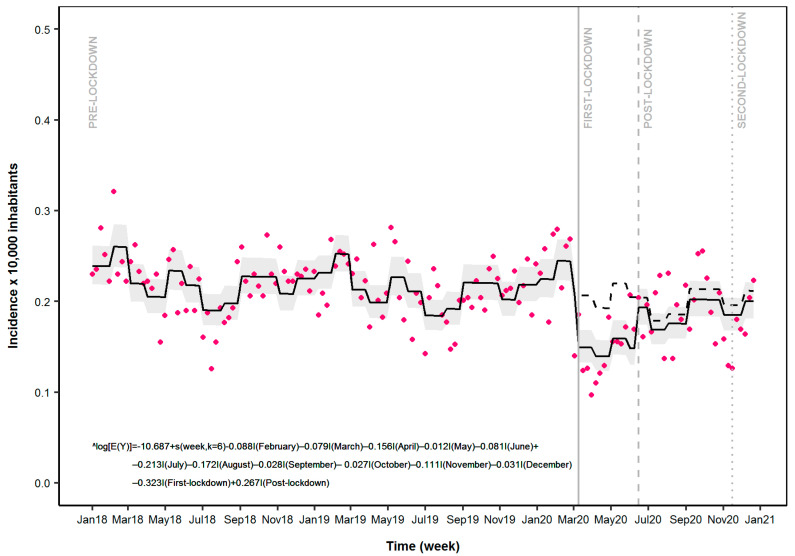
Time series analysis of emergency department access across different periods: Pre-Lockdown, First Lockdown, Post-Lockdown and Second Lockdown. Legend: Magenta dots: estimated incidence of Emergency department access; solid line: predicted model based on estimated outcome; Grey zone: 95%CI of the predicted model; Dashed line: expected scenario in the absence of LMI.

**Table 1 ijerph-18-13253-t001:** Time series analysis of anti-seizure medication (ASM) use and emergency department access across different time segments: Pre-Lockdown, First Lockdown, Post-Lockdown and Second Lockdown.

Prevalence of ASM Use				
**Model Parameter ^1^**	**β**	**Prevalence Ratio**	**95%CI**	***p*-Value**
First Lockdown (4th week) ^3^	−0.054	0.947	0.935–0.960	<0.001
Second Lockdown (3rd week) ^4^	−0.030	0.970	0.962–0.978	<0.001
**Incidence of ASM Use**				
**Model parameter1**	**β**	**Incidence Ratio**	**95%CI**	***p*-value**
First Lockdown (1st week) ^3^	−0.426	0.653	0.593–0.719	<0.001
First Lockdown (4th week) ^4^	0.036	1.037	1.027–1.046	<0.001
Post-Lockdown (4th week) ^4^	−0.033	0.967	0.956–0.979	<0.001
Second Lockdown (1st week) ^3^	−0.118	0.889	0.799–0.990	0.032
**Emergency Department access**				
**Model parameter ^2^**	**β**	**Events Ratio**	**95%CI**	***p*-value**
First Lockdown (1st week) ^3^	−0.323	0.724	0.638–0.822	<0.001
Post-First Lockdown (1st week) ^4^	0.267	1.306	1.150–1.484	<0.001

^1^ The GAM model was also corrected for holiday and a spline function of week (k = 6). ^2^ The GAM model was also corrected for month and a spline function of week (k = 6). ^3^ Level change; ^4^ Slope change.

## Data Availability

Data sharing is not applicable to this article.

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
