# Peer review of "Impact of COVID-19 Lockdown, during the Two Waves, on Drug Use and Emergency Department Access in People with Epilepsy: An Interrupted Time-Series Analysis"

_ijerph, 2021, doi:10.3390/ijerph182413253_

Round 1

Reviewer 1 Report

The present study aims to evaluate the impact of COVID-19 on the incidence and prevalence of ASM and ED access in patients with epilepsy.

The study is well-performed and address an important issue that has been under investigated.

I suggest only few elements to improve the manuscript:

  • In the limitations paragraph it is important to underline that these data represents only a small sample of the entire population of Italy. Many differences in the healthcare organization during the pandemic could have occurred. Consequently, any generalization should be avoided.
  • The R package used should be reported in the statistical analysis paragraph
  • Adequate considerations in the discussion should be reported for patients newly diagnosed with epilepsy.   

Author Response

Reviewer 1

The present study aims to evaluate the impact of COVID-19 on the incidence and prevalence of ASM and ED access in patients with epilepsy.

The study is well-performed and address an important issue that has been under investigated.

I suggest only few elements to improve the manuscript:

We thank the reviewer for the comments to our manuscript. In the new version of the manuscript we carefully considered all raised points. 

Our point-by-point responses are provided below. 

  • In the limitations paragraph it is important to underline that these data represents only a small sample of the entire population of Italy. Many differences in the healthcare organization during the pandemic could have occurred. Consequently, any generalization should be avoided.

We thank the reviewer for the comment. In the last version of the manuscript, we have updated the limitations paragraph by including the following sentence: “…Finally, regional differences could limit the generalization of our findings to the whole Italian population. Moreover, different lockdown measures implemented during the two COVID-19 waves across the Italian regions and European countries might have impacted differently on the drug use and healthcare resources utilization. Further researches should explore the impact of the restrictive public health measures in European and non-European countries on the aforementioned outcomes.”  (pages 7-8, lines 252-267). 

  • The R package used should be reported in the statistical analysis paragraph

We agree with the reviewer. In the method section we added the following sentence “In this study both data processing and data analysis were performed using the R studio software (version 4.0.2). In particular, the ITS analysis was conducted by using “tsMODEL”, “splines”, and “mgcv” R packages”. (page 4, lines 154-155)

  • Adequate considerations in the discussion should be reported for patients newly diagnosed with epilepsy.   

We thank the reviewer for the comment. In the new version of the manuscript, we have update the discussion section by adding more details on potential reasons for the observed increase in incidence of ASMs use after post-lockdown. “….  In addition, it is worth mentioning some evidence that suggested a neurotropism of SARS-CoV-2 with consequent onset of neurological symptoms and potential sequelae in COVID-19 patients such as dizziness, ataxia and seizures. Therefore, it is reasonable to speculate that a minority of the new ASMs users might be treated for epilepsy and to prevent further seizures resulting from post-acute COVID-19 [41, 42].”  (page 7, lines 232-237).

References

  1. Asadi-Pooya AA. Seizures associated with coronavirus infections. Seizure. 2020 Jul;79:49-52. doi: 10.1016/j.seizure.2020.05.005..
  2. Asadi-Pooya AA, Simani L, Shahisavandi M, Barzegar Z. COVID-19, de novo seizures, and epilepsy: a systematic review. Neurol Sci. 2021 Feb;42(2):415-431. doi: 10.1007/s10072-020-04932-2

Reviewer 2 Report

I read with interest the study of Antonazzo and colleagues on the effects of COVID-19 lockdowns among people with epilepsy in terms of drug use and ED access using real time data and found increased incidence of ASM use and ED access. I have several concerns however.

Major:

  1. The authors need to justify the use of interrupted time series to answer their objectives as it needs to fulfill 3 assumptions to have any credence. The assumption of linearity in this case can be very hard to validate since they used very few time points. The objectives can be answered using simpler statistics.
  2. There was no description of the subjects involved at all as well the sample size.
  3. More detailed description of the subjects including inclusion and exclusion criteria.
  4. The authors need to elaborate more on the novelty and significance of the study as the observed results may variably apply to any population with pre-existing chronic disease during lockdown.

Minor

  1. The title is misleading since its not the outbreak per se and its relation to changes in ASM use and ED access being observed but rather the lockdowns imposed.
  2. The authors can remove the word 'non-interventional' it will appear redundant since observational was already mentioned.
  3. IRB number and date of approval should be provided if available.

Author Response

Reviewer 2

I read with interest the study of Antonazzo and colleagues on the effects of COVID-19 lockdowns among people with epilepsy in terms of drug use and ED access using real time data and found increased incidence of ASM use and ED access. I have several concerns however.

Thank you for the opportunity to address the reviewer’s comments, which we have carefully considered when preparing the revised version of the manuscript.

Our point-by-point responses are provided below. 

Major:

  • The authors need to justify the use of interrupted time series to answer their objectives as it needs to fulfill 3 assumptions to have any credence. The assumption of linearity in this case can be very hard to validate since they used very few time points. The objectives can be answered using simpler statistics.

Thank you for your comment. It allows us to be clearer in relation to our methodological choices. We decided to use an interrupted-time series approach because it is recognized as a robust quasi-experimental design to evaluate longitudinal effects of time-delimited interventions (Kontopantelis 2015, Wagneer 2005, Pehnfold 2020) on a population level outcome. This method allowed us to compare longitudinal outcomes among time periods accounting for different time lengths and for secular time trends. These two aspects could influence a simple basic comparison analysis among periods.

Moreover, we used generalize additive models to overcome the assumption of linearity. Using these kind of model, we could adjust for seasonality and non-linear secular time trends, reducing the autocorrelation between observation, which are typical features of time-series.

In the new version of manuscript, we added a sentence in the methods section to clarify our choice: “We then used an Interrupted Time Series (ITS) approach [20, 21] to estimate the effect of LM on observed outcomes time series. This is a quasi-experimental design to evaluate longitudinal effects of time-delimited interventions accounting for seasonality and secular trend. In detail, we applied a quasi-Poisson generalized additive model [22] with the weekly count of the observed outcome as response variable (Y) and the reference population as offset variable to transform the count outcome in incidence or prevalence.” (page 3, lines 125-128).

Penfold R and Zhang F. Use of interrupted Time Series Analysis in Evaluatng Helath Care Quality Improvements.  Acdemic Pediartric 2020( 13):S38.

Kontopantelis E. Regression base quasi-experimentale approache when randomisation is not an option:interrupted tiem series analysis.  BMJ 2015; 350

Wagner AK. Segemented regression analysis of interrupted time series studies in medication use reserach. Journ of Clinicla Pharmacy and Therapeutics 2002;27 :299.

  • There was no description of the subjects involved at all as well the sample size.

We thank the reviewer for the comment which give use the possibility to add more data to the manuscript. In the new version of methods section, we included the following sentence “The population was stable through the study period, males accounted for the 48%, 15% of the population was under 18-years and 32% was above 60 years.”. (page 2, lines 87-89)

As regarding the description of included individuals, we didn’t include a description of the number of subjects with the outcome of interests among segments of the study period, because segments have different time length affecting the number of patients observed and we thought this could be misleading for the reader.

  • More detailed description of the subjects including inclusion and exclusion criteria.

In the new version of the method we have rephrased the exclusion criteria sentience as it follows “We excluded patients with less than 1 year of database history prior to the index”. (page 3, lines 94-95)

  • The authors need to elaborate more on the novelty and significance of the study as the observed results may variably apply to any population with pre-existing chronic disease during lockdown.

Thank you for the comment. In the new version of conclusion paragraph, we have added more details on this aspect by including the following sentence “…Finally, from methodological point of view this type of analysis represents a reliable method to explore whether similar trends might have occurred among individuals with other chronic conditions during the emergency situation. These results might be used by relevant stakeholders to implement new strategies to guarantee continued medical care in future public health emergencies. Additionally, evidence from these studies might be used to target specific health intervention in sub-group of patients who have peculiar needed to mitigate the impact of future COVID-19 waves on their management”. (page 8, lines 267-274)

Minor

  1. The title is misleading since its not the outbreak per se and its relation to changes in ASM use and ED access being observed but rather the lockdowns imposed.

We agree with the reviewer. Therefore, we have changed the title in “Impact of COVID-19 lockdown, during the two waves, on Drug Use and Emergency Department Access in People with Epilepsy: An Interrupted Time-series Analysis”.  (page 1)

  1. The authors can remove the word 'non-interventional' it will appear redundant since observational was already mentioned.

We have removed the word non-interventional.  (page 2, line 71)

  1. IRB number and date of approval should be provided if available.

Thank you for the comments, in the manuscript is missing the IRB number as well as the date of approval because the “Agenzia regionale di Sanità della Toscana” has general legal and ethical frameworks that allow conducting research by making secondary use of administrative data. However, we would like to specify that the study was conducted according with the “European Network of Centres for Pharmacoepidemiology and Pharmacovigilance” guideline.

Round 2

Reviewer 2 Report

I am satisfied with the authors' rebuttal.